# Bimetallic Nanomaterials: A Promising Nanoplatform for Multimodal Cancer Therapy

**DOI:** 10.3390/molecules27248712

**Published:** 2022-12-09

**Authors:** Guiming Niu, Fucheng Gao, Yandong Wang, Jie Zhang, Li Zhao, Yanyan Jiang

**Affiliations:** Key Laboratory for Liquid−Solid Structural Evolution and Processing of Materials, Ministry of Education, Shandong University, Jinan 250061, China

**Keywords:** bimetallic nanomaterials, synthesis methods, physicochemical properties, multimodal cancer therapy

## Abstract

Bimetallic nanomaterials (BMNs) composed of two different metal elements have certain mixing patterns and geometric structures, and they often have superior properties than monometallic nanomaterials. Bimetallic-based nanomaterials have been widely investigated and extensively used in many biomedical fields especially cancer therapy because of their unique morphology and structure, special physicochemical properties, excellent biocompatibility, and synergistic effect. However, most reviews focused on the application of BMNs in cancer diagnoses (sensing, and imaging) and rarely mentioned the application of the treatment of cancer. The purpose of this review is to provide a comprehensive perspective on the recent progress of BNMs as therapeutic agents. We first introduce and discuss the synthesis methods, intrinsic properties (size, morphology, and structure), and optical and catalytic properties relevant to cancer therapy. Then, we highlight the application of BMNs in cancer therapy (e.g., drug/gene delivery, radiotherapy, photothermal therapy, photodynamic therapy, enzyme-mediated tumor therapy, and multifunctional synergistic therapy). Finally, we put forward insights for the forthcoming in order to make more comprehensive use of BMNs and improve the medical system of cancer treatment.

## 1. Introduction

In recent years, metallic nanomaterials (MNs) have made remarkable achievements in the biomedical field due to their unique optical properties, nano−size effects, and good biocompatibility [1,2,3,4]. Among them, MNs-mediated nanotechnology has attracted a lot of attention to achievements in tumor therapy and solved the side effects, trauma, and incomplete treatment in traditional cancer treatment, such as photothermal therapy (PTT) [5,6], photodynamic therapy (PDT) [7], gene therapy (GT) [8], and so on. Interestingly, traditional alloys have been created to obtain better stability and mechanical properties than mono metals, as well as when scaled down to the nanoscale. It has been shown that bimetallic nanomaterials (BMNs) perform similar or even better physical and chemical properties compared with monometallic nanomaterials (MMNs), their excellent photothermal and photocatalytic properties and enzyme-like activity make them more suitable for tumor therapy [9]. Therefore, BMNs have attracted extensive attention in the medical field.

In this review, BMNs refer to nano-bimetallic alloys, intermetallic compounds, or the combination of two kinds of metallic nanoparticles (NPs). BMNs with surface plasmon resonance (SPR) effect have been used in cancer therapy due to their excellent optical properties. A typical example is the Pt nanoparticle, a good photothermal agent and enzyme-like catalyst [10], that performs cascade catalytic activity and more potent therapeutic effects when combined with another catalytic metal nanoparticle (NP) [11,12]. As demonstrated by previous studies, compared with MMNs, BMNs have good reusability in heating/cooling cycles and can realize the adjustment of photothermal conversion efficiency more accurately by adjusting the morphology, which solves the problem of low photothermal efficiency in PTT with MMNs [13]. Meanwhile, due to their unique physical and chemical properties, especially surface chemistry, BMNs have better biocompatibility, drug loading ability, and radiosensitization ability, which provides a great help in solving the limitations of MMNs-mediated cancer therapy [14,15]. In addition, how to realize the advantages of BMNs for tumor therapy is critical.

Generally, doping with the second metal is considered to possess superior performance than MMNs, due to the addition of more active sites by the formation of metal polar bonds and irregular arrangement [16,17]. At the same time, the combination of the two metals is more likely to form complex structures with enhanced SPR effects such as hollow structures, porous structures, and core–shell structures [18]. Therefore, regulating the optical, electrical, chemical, and biological properties by controlling their size, shape, and composition during synthesis is particularly important in the study of BMNs [19]. In order to achieve the design and effects, many methods have been extensively tested in the past decades. Among them, there is an important problem that the ratio of the two metals and the shape of BMNs are difficult to control [20]. BMNs with controlled composition can be synthesized by the co-reduction method and hydrothermal method, and their shapes are mostly spherical and massive. BMNs synthesized by the seed-mediated method usually perform complex shapes, while the BMNs obtained by electrodeposition are mostly nanowires and nanofilms. In order to obtain suitable BMNs for cancer therapy, a systematic understanding of the synthesis methods is required.

Numerous studies have shown that BMNs have been widely used in bioimaging, biosensing, and tumor treatment due to their diverse structures and excellent physicochemical properties. To date, most reviews have focused on the regulation of molecular composition, crystal structure, and physicochemical properties, or concentrated on the applications in sensing and imaging, while little attention has been paid to BMNs in cancer therapy [21]. In view of the promising development prospects of BMNs, this review summarizes and discusses the synthesis methods, physicochemical properties, and applications of BMNs related to cancer therapy. Ultimately, taking BMNs suitable for tumor therapy as the point, some insights for the investigation including the existing problems and prospects for future research on BMNs are brought forward. This review aims to provide a convenient way for researchers to understand the application of BMNs in cancer therapy and provide guidance or strategies for future research.

## 2. Synthesis Methods of BMNs

Many available methods for the synthesis of BMNs are reported, while some products can be generated by different methods due to similarities in basic mechanics [22,23]. In this review, we summarized the preparation methods of BMNs commonly used for cancer therapy, such as the co-reduction method, hydrothermal method, seed-mediated growth method, and electrodeposition method.

### 2.1. Co-Reduction Method

Co-reduction is one of the most straightforward and convenient methods to prepare BMNs due to its simple operation, low cost, and short reaction time. Co-reduction, also known as the one-pot method, is usually used when two precursors containing metal elements are mixed, and the metal ions are reduced to form alloys or intermetallic compounds [24,25]. In addition, the BMNs’ morphology and structure can be tailored by the reaction temperature, the addition of surfactants, the reducing agent, and the nature of the coordination ligand. Many common BMNs such as gold/silver core–shell structures, gold/silver nanowires, and gold/platinum NPs are synthesized by this method. For example, polyvinylpyrrolidone-platinum-copper nanoparticle clusters (PVP-PtCuNCs) were developed by a facile method by Liu et al. [26]. In their study, PVP-PtCuNCs were reduced from H_2_PtCl_6_·6H_2_O solution and CuSO_4_·5H_2_O solution by ascorbic acid solution and then modified with PVP. The as-synthesized PVP-PtCuNCs exhibited excellent multiple enzyme-mimicking activities, such as peroxidase (POD)-like, catalase (CAT)-like, and superoxide dismutase (SOD)-like activities and high ·OH-scavenging ability.

More sophisticated nanomaterials with special shapes and structures can be synthesized by co-reduction in combination with other methods. Joo et al. prepared monodispersed Ag cubic to mesh nanostructures embedded with Au nanoparticles (AgCM/AuNPs) using co-reduction and template-assisted synthesis [27]. The AgCM/AuNPs were composed of six equivalent panels with similar nanostructures, thus possessing highly attractive plasmonic photocatalysts owing to the large surface area based on the meshed nanostructures (Figure 1a). Takeuchi et al. presented a combination of galvanic replacement reaction with a co-reducing agent to synthesize a metallic NPs of Ag-Pt double-shell on Au-core (Au@Ag@Pt core@multishell) with a hollow-granular shell structure [28].

### 2.2. Hydrothermal Method

Another widely used method is the hydrothermal method, which is similar to the co-reduction method. The metal precursors are promoted to decompose and reduce after heating. The hydrothermal method is generally applicable to reactions with lower reduction potentials that are not easily reduced directly. Up to date, some BMNs synthesized from metal precursors with lower reduction potentials have been prepared with this method, such as CuNi [29,30], Ni-Fe [31], CoNi [32], and NiRe [33]. For example, Gai et al. designed and prepared NiFe alloy NPs only using hydrochar, Ni(II) nitrate nonahydrate, and Fe(III) nitrate nonahydrate, as shown in Figure 1b. The as-synthesized NiFe NP has adjustable morphology size and catalytic properties. Besides, the hydrothermal method is not only suitable for metal precursors with low reducibility but also widely used in the synthesis of many BMNs (PtCu, PtPd, et al., for instance) [34,35]. Wang and Yin’s groups investigated bimetallic CuAu_x_ (x = 0.01–0.04) NP catalysts [36]. The NPs in CuAu_x_ were around 13 and 5 nm. Liu et al. developed a bimetallic PdCu NP decorated three-dimensional graphene hydrogel (PdCu/GE) by a simple hydrothermal method [37]. In a word, the hydrothermal method is simple to operate and easy to synthesize large quantities of BMNs.

### 2.3. Seed-Mediated Growth Method

Seed-mediated growth has been proven an effective method for synthesizing plasmonic noble metal nanocrystals [38]. This method has been applied in the synthesis of BMNs because the prepared nanocrystals have well-defined morphology, size, and surface composition. Typically, seed-mediated methods play an important role in the preparation of anisotropic metal structures and hierarchical epitaxial core/shell structures [39,40,41,42]. Xia et al. reported a facile route to the synthesis of Au@Ag core–shell nanocubes in the aqueous solution [43]. First, very small (2–3 nm) Au nanocrystallites were synthesized by reducing HAuCl_4_ with NaBH_4_ in the solution of cetyltrimethylammonium bromide (CTAB). Then, HAuCl_4_, cetyltrimethylammonium chloride (CTAC), and aqueous l-ascorbic acid (AA) were dissolved in ultrapure water, followed by the addition of the 3 nm Au NPs. The mixture turned from colorless into red, indicating the formation of larger Au seeds. Whereafter, CTAC-Au seeds and CTAC aqueous solution were mixed and heated under magnetic stirring. Finally, a specific volume of aqueous AgNO_3_ solution and an aqueous solution of AA and CTAC were simultaneously injected to form the Au@Ag nanocubes. In this work, the thickness of the Ag shell and the size of Au@Ag could be precisely adjusted, and the method is supposed to be environmentally friendlier than other methods.

In addition to Au-Ag BMNs, some other novel metal BMNs with high activity can also be easily synthesized by this method. Cargnello’s group developed a seed-mediated approach to synthesizing monodisperse Pt_x_Cu_100-x_ nanocrystal alloys with controlled Pt/Cu ratios [44]. They demonstrated that Pt and Cu in individual nanocrystals were successfully alloyed as well as Pt_x_Cu_100-x_ with controlled composition. Seed-mediated growth method has emerged in the field of synthesizing BMNs due to simple operation, strong repeatability, and high yield. In particular, the shapes and components of BMNs can be accurately regulated by the seed-mediated method, which is difficult to obtain by other methods. It is believed that this method will be more extensively used in the future. 

### 2.4. Electrodeposition Method

Electrodeposition is a simple and inexpensive synthesis method, which is widely used in the synthesis of nanotube, nanoporosity, and nanofilm [45,46,47]. For example, an innovative technology of striped gold–silver alloy nanowires (Au–Ag alloy NWs) was presented by Karn-orachai et al. (Figure 1c) [48]. In this work, the NWs were prepared by electrodeposition of Au–Ag alloy into the pores of etched ion-track membranes. The Au–Ag alloy NWs with a high surface-to-volume ratio is obtained by electrodeposition, which combines the advantages of porous Au and 1D nanostructure and has broad application prospects. In addition, Jia et al. synthesized Au-Pt nanostructures containing nanoporous Au and Pt nanoparticles by electrodeposition [49]. Briefly, nanoporous Au (NPG) was first deposited on an electrode by dealloying of commercial alloy film, and then Pt nanoparticles were decorated on NPG by electrodeposition. In this work, the Au–Pt nanostructures were constructed to be a label-free immunosensor by directly immobilizing the antibody of NMP22 on the surface. Meanwhile, the BMNs showed electrocatalytic activity in the reduction of H_2_O_2_. The electrodeposition method also can synthesize other MNs. Downard et al. used copper foil as a carrier to form Cu and Au NPs by electrodeposition to modify graphene [50]. The hierarchical nanostructure of various metals can also be synthesized by electrodeposition. Kim and coworkers presented a vertical growth of Ni-Cu-Se nanoflakes via the co-electrodeposition technique [51]. This method is rarely used to produce BMNs for disease treatment, because of its limitations including harsh reaction conditions, low yield, and difficulty to control.

**Figure 1 molecules-27-08712-f001:**
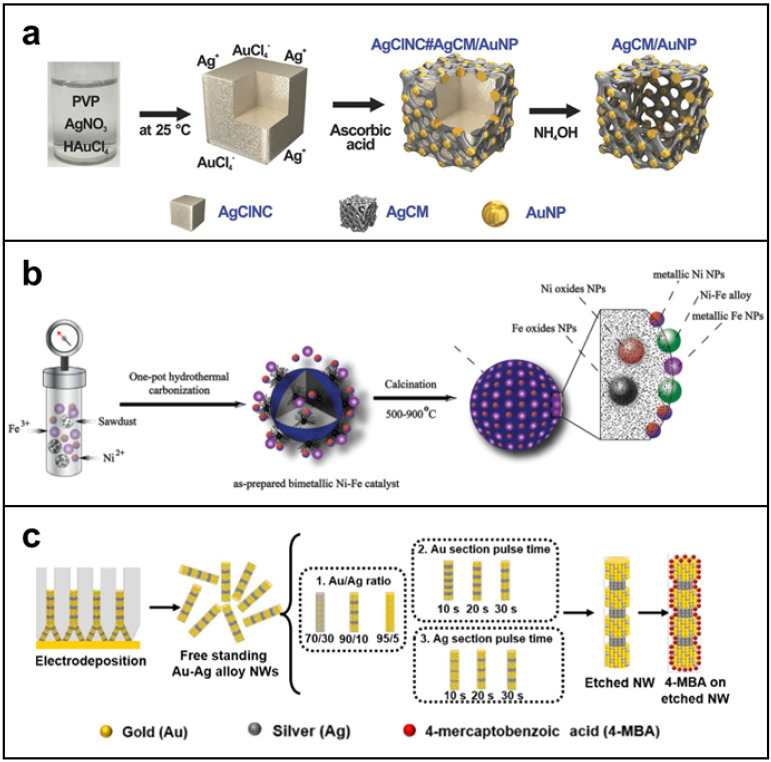
The synthesis methods of BMNs. (**a**) Co-reduction and template-assisted synthesis used for AgCM/AuNPs. Adapted with permission from Ref. [27]. Copyright © 2017 John Wiley & Sons, Inc. (**b**) Schematic diagram of hydrothermal synthesis of NiFe alloy NPs. Reprinted with permission from Ref. [31]. Copyright © Royal Society of Chemistry 2022. (**c**) The principle of electrodeposition for the synthesis of Au–Ag alloy nanowires. Reprinted with permission from Ref. [48]. Copyright © 2019 Elsevier B.V.

In conclusion, the co-reduction method, hydrothermal method, and seed-mediated method have been widely used for the preparation of BMNs suitable for cancer therapy. Especially the seed-mediated method can precisely control the morphology of BMNs to obtain unparalleled properties. Although precise porous BMNs can also be obtained by the electrodeposition method and seed-mediated method, which are difficult to be obtained by other methods. It is exciting that BMNs with delicate structures can be synthesized more conveniently with the assistance of two synthesis methods, such as seed-mediated method and co-reduction [52], template synthesis and co-reduction [27]. Therefore, it is very important to choose the appropriate method according to the properties and morphology of BMNs.

## 3. Unique Properties of BMNs Relevant to Cancer Therapy

### 3.1. Intrinsic Properties of BMNs

It is well known that many nanoparticles have excellent properties due to their unique morphological characteristics and structure. Common nanomaterials used in disease treatment, such as metallic nanoparticles, polymer nanoparticles, and inorganic nanoparticles, are mainly manifested as nanospheres, nanorods, nanoclusters, nanostars, and nanoflowers, which have large surface areas and unique optical properties [53,54,55,56,57,58]. Similarly, BMNs also exhibit various and complex morphologies based on spatial arrangements, which are related to abundant physicochemical properties (Figure 2).

Among them, dendritic-like structures endowed with BMNs are more active sites because of the steps, edges, and corners. For example, Zhou and co-workers prepared the adenosine 5’-triphosphate @Au-Cu nanoparticle (ATP@Au-Cu NP) with a multiarm shape structure to destroy the membrane [59]. As we can see from Figure 2a, ATP@Au–Cu NP performs multiarm shape, successful doping of the bimetal, and a good crystal structure. As such, the Au–Pt nanodendrites (Au–Pt NDs) with the face-centered cubic (FCC) structure were created to enhance the electronic interaction to obtain superior catalytic activity (Figure 2c) [60]. Kohane et al. developed a multifunctional NP, which is a gold nanostar (AuNS) core coated with a metal-drug coordination polymer (CP) shell, as shown in Figure 2d [61]. The nanosystem (AuNS@CP) is considered a good photothermal agent for PTT and a vehicle to deliver therapeutics due to retaining the original shape of the AuNS core with SPR and two-photon luminescence (TPL). In addition, many novel BMNs with exquisite shapes have been created, such as Au@Pt nanodendrites (Au@Pt NDs) and Cu–Pd alloy tetrapod nanoparticles (TNP), as exhibited in Figure 2b,e [62,63]. All of them show great potential in cancer treatment because of their high catalytic activity, good photothermal effect, remarkable biocompatibility, and drug-loading capacity.

Meanwhile, BMNs with alloy and nanocrystalline structures usually perform a wide range of applications in cancer treatment. Paul’s group reported the Ag–Au alloy nanoparticle with FCC structure as a drug delivery agent for cancer in the pharmaceutical application (Figure 2f) [19]. Decorating with another metal to form baroque morphologies is a common way to improve the effectiveness of anticancer therapy, as shown in Figure 2g,h [64,65]. As for nanorods, a common monometallic nanostructure can also be made in this way (Figure 2i). Liu et al. presented Au@Pt nanorods that were composed of single crystalline Pt nanodots covered on Au NRs [66]. The photothermal conversions were enhanced and the cytotoxicity was significantly decreased. In Pavithra’s study, a bimetallic nanofusiform-like structure doped with carbon nanodots (Co/Mn@CNDs-MOF) was prepared by solvothermal treatment (Figure 2j) [67]. What is more, the BMNs can also be distributed as NWs, and their proportions and sizes are controllable to regulate ferromagnetism [68].

Core–shell structure is universal among the BMNs used in cancer treatment, because of its unique optical properties, as well as complex surface shape and porous nature. The most commonly used are Ag–Au alloy and Ag@Au core–shell type NPs (Figure 2k) [69]. In the study, different anticancer effects can be achieved by adjusting the ratio of Au–Ag and the shape of the core–shell structure.

Moreover, core–shell nanostructures can coexist with porous structures. For example, Wang et al. reported a porous Au@Rh bimetallic core–shell nanostructure [70]. As shown in Figure 2l, the Au core and porous Rh shell exhibited strong broadband absorption and superior catalytic activity. Besides, porous nanostructures have more active sites because of their large specific surface area and the abundance of atomic steps, edges, and corner atoms, thus widely used in tumor therapy. As proven in Figure 2m, the cubic mesh nanostructures of AgCM/AuNPs enhanced high surface area and photocatalytic activity and performed a highly elaborate and facile design of nanomaterials [27].

**Figure 2 molecules-27-08712-f002:**
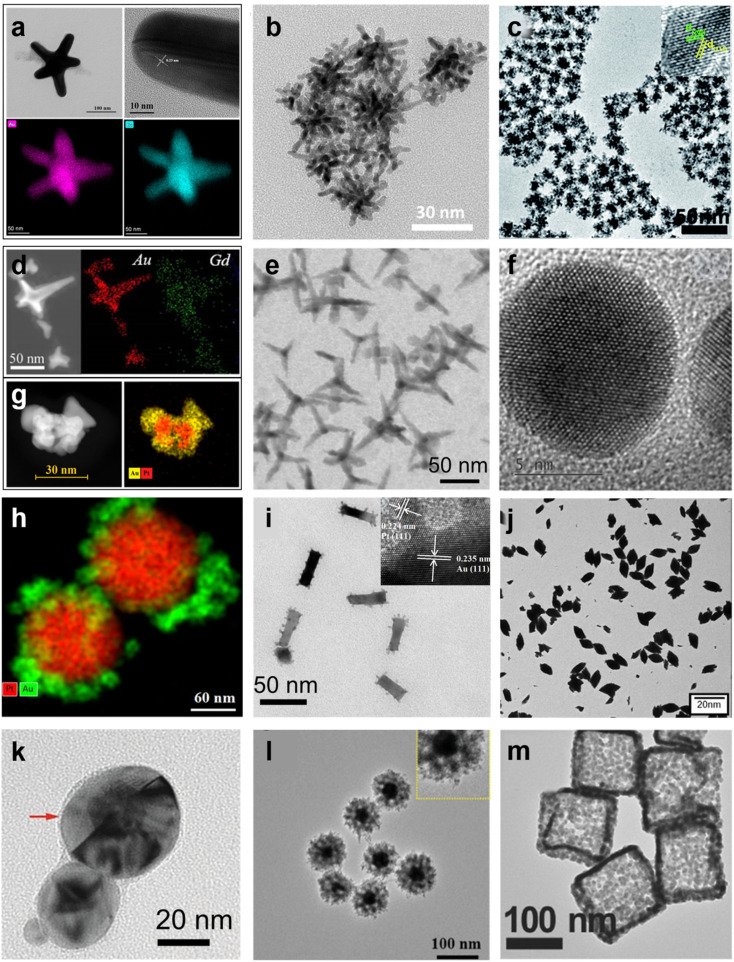
Morphologies and structures of different BMNs. (**a**) Characterization of ATP@Au–Cu NP (TEM, HR-TEM, and EDS). Reprinted with permission from Ref. [59]. Copyright © 2022 American Chemical Society. (**b**) TEM image of Au@Pt NDs. Reprinted with permission from Ref. [62]. Copyright © 2016 American Chemical Society (**c**) TEM image and HR-TEM of Au-Pt NDs. Adapted with permission from Ref. [60]. Copyright © The Royal Society of Chemistry 2015. (**d**) The TEM image and EDS images of AuNS@CP (space between black and white lines indicates CP layer). Reprinted with permission from Ref. [61]. Copyright ©2022 Ivyspring International Publisher. (**e**) TEM image of Cu–Pd alloy TNP. Reprinted with permission from Ref. [63]. Copyright © 2022 Springer Nature Limited. (**f**) HR-TEM of Ag–Au alloy NP. Reprinted with permission from Ref. [19]. Copyright © 2019 Elsevier B.V. (**g**) TEM image and EDS image of gold-decorated platinum NPs. Reprinted with permission from Ref. [64]. Copyright © 2021 MDPI. (**h**) EDS image of platinum–gold nanoraspberries. Reprinted with permission from Ref. [65]. Copyright © 2019 Springer Nature. (**i**) TEM image of Au@Pt0.05. Adapted with permission from Ref. [66]. Copyright © 2017 Elsevier B.V. (**j**) TEM image of Co/Mn@CNDs-MOF. Reprinted with permission from Ref. [67]. Copyright © 2022 American Chemical Society. (**k**) TEM of NanoGS (Au@Ag NPs). Reprinted with permission from Ref. [69]. Copyright © 2022 Elsevier Ltd. (**l**) TEM of porous Au@Rh bimetallic core–shell nanostructure. Reprinted with permission from Ref. [70]. Copyright © 2020 John Wiley & Sons, Inc. (**m**) TEM of AgCM/AuNPs. Reprinted with permission from Ref. [27]. Copyright © 2017 John Wiley & Sons, Inc.

In summary, many complex and diverse morphologies can be exhibited in BMNs that indicate the suitable characteristics for tumor therapy, such as good biocompatibility, drug-loading capacity, and photothermal and catalytic properties. BMNs have a broad development prospect in tumor therapy by virtue of these abundant morphologies.

### 3.2. Optical Properties

BMNs, such as gold and silver NPs, have excellent optical properties due to their localized surface plasmon resonance (LSPR) properties. Meanwhile, BMNs have attracted much attention because of their visible-to-near-infrared tunable plasmonic features, high chemical stability, and so on [71]. The LSPR of BMNs caused by that conduction electrons coherent oscillates in response to incoming light and then released heat or energy, has been widely used in photothermal conversion, photocatalysis, and biosensors [72]. Furthermore, the optical properties of BMNs are affected by their structure and shape, so changing the shape and size of NPs is an effective measure to adjust the optical properties, which provides more diversities in the applications of BMNs.

#### 3.2.1. Photothermal Properties

BMNs usually have a satisfactory photothermal effect because the original resonance effect can be improved when integrating two metallic nanomaterials with significant LSPR, such as Au, Ag, Pt, and Cu, in crystallization or physical means.

In a typical PTT, to ensure maximum tissue penetration and avoid injury to normal tissues, 808 nm wavelength lasers located in the near-infrared (NIR) region are usually selected for treatment [73]. Generally, the LSPR of nanomaterials can be generated and the photothermal performance is better when their absorbance at laser wavelength is stronger. BMNs have excellent LSPR due to their compositions, special shape, and nano scale. At the same time, LSPR of BMNs can be controlled by adjusting the component and shape, so as to obtain a better photothermal effect, as shown in Figure 3. In Sang’s work, dumbbell-shaped Au–Pt bimetallic nanorods (AuPtNRs) have been formed by depositing Pt on the tips of gold nanorods (AuNRs) [74]. The UV absorbance in Figure 3a shows that the longitudinal LSPR peak red-shifted to a longer wavelength with the Pt primarily depositing at the tips of AuNRs and forming dumbbell−shaped AuPtNRs. In this way, the absorption peak is closer to 808 nm, resulting in better photothermal conversion and facilitating PTT. After 10 min of laser irradiation, the temperature of AuNRs solution increased by only 27 °C from the initial temperature to 53.6 °C. In contrast, the AuPtNRs solution rapidly heated up to 63.1 °C, reflecting a more intense photothermal effect. This is attributed to their LSPR peaks being closer to 808 nm and having a wider spectral band, which enables them to absorb more photon energy and release more heat.

For BMNs without an obvious absorption peak of LSPR, the absorbance is larger, the more energy is absorbed under the same light, and the more heat is released for PTT. Pan et al. designed PEGylated Au@Pt NDs as a multifunctional and synergistic agent for cancer therapy [62]. The absorption of Au@Pt NDs was switched to the NIR region with the growth of Pt nanobranches, thus enhancing the efficacy of PTT. There is no obvious absorption peak in the spectrum, so the absorbance of Au@Pt NDs solution with a certain concentration gradient is used to prove the LSPR of Au@Pt NDs and the increasing photothermal effects, as depicted in Figure 3b. In another study, a type of Cu–Pd alloy tetrapod NPs (TNP-1) exhibited superior NIR photothermal conversion [63]. By comparing three types of Cu–Pd alloy NPs, including TNP-1, TNP-2, and spherical nanoparticle (SNP), in the ultraviolet absorption spectrum, the absorbance of TNP-2 is highest and that of TNP-1 is higher than SNP; correspondingly, in the test of photothermal effect, TNPs had markedly enhanced photothermal conversion efficiency (Figure 3c). It is reasonable that the sharp-tip structure of TNPs could concentrate light at their tips and thus promote more outstanding PTT. 

#### 3.2.2. Photocatalytic Properties

In the bimetallic system, the LSPR and catalytic performance of metal materials are combined to promote each other, showing an attractive photocatalytic effect with laser irradiation. Energetic charge carriers are generated on the surface of optically excited plasmonic NPs and then arrive at the adsorbed substance, where they reduce the activation energy of the reaction and play a catalytic role [72].

BMNs containing catalytic metals (e.g., Pt, Pd, and Cu) and metals with SPR properties (e.g., Au and Ag) usually have good photocatalytic properties [75,76]. The common combinations of these metals, the products, and the excitation wavelengths required are shown in Table 1. In the bimetallic nanosystems, BMNs always act as photosensitizers to absorb the energy of NIR laser; therefore, it can produce holes (h^+^) and photoinduced electrons (e^−^) pairs, where h^+^ is a powerful oxidant and e^−^ is a strong reductant [71,72]. A typical example of a reaction induced by excited h^+^ and e^−^ pair is the production of H_2_ [77]. The photocatalytic reduction of Pt-modified Au NRs has been shown in Figure 4a, under NIR laser irradiation, the hot electrons generated either by LSPR excitation or interband excitation transfer to the tip-coated Pt, where H^+^ reduces to H_2_. The spatial separation of reduction and oxidation sites in Pt-tipped Au NRs results in an efficient charge separation (CR) and more production of H_2_. What is more, the generation of reactive oxygen species (ROS) such as superoxide anion (^•^O_2_^−^), singlet oxygen (O^•−^), hydroxyl radical (^•^OH), and hydrogen peroxide (H_2_O_2_) under laser irradiation is a brilliant application [78]. For example, novel silver–copper bimetallic alloy (Ag–Cu) NPs decorated β-NaYF4: Yb^3+^, Tm^3+^ @TiO_2_ (“@” means TiO_2_ supported on NaYF4: Yb^3+^, Tm^3+^, marked as NYFT) microrods successfully promote formation rate of e^−^ and h^+^ due to their SPR effect, thus generating reactive radical ^•^O_2_^−^ and degrading Rhodamine B (RhB) to intermediate or CO_2_ and H_2_O, as shown in Figure 4b. The energy level transitions from ground state to excited state after the sensitizer ion Yb^3+^ absorbing NIR laser (980 nm), then energy would be successively transferred to adjacent other higher energy levels. Therefore, it can absorb three wavelengths of laser (291, 347, and 363 nm) and produce e^−^ and h^+^. In addition, photoinduced electrons are easily injected into Ag–Cu alloy particles, and Ag donates e^−^ to Cu. Then e^−^ abundantly reserved in Cu NPs are trapped by O_2_ to generate reactive radical ^•^O_2_^−^ with the SPR.

### 3.3. Enzyme-Like Activity

BMNs have also attracted much attention because of their catalytic activity, which is known as an enzyme-like activity when they are used in medicine [88]. Many studies have shown that the BMNs exhibited higher catalytic activities than MMNs due to the synergistic and electron transfer effect; and its enzyme-like activity is responsive to external stimuli such as pH [89], temperature [90], laser [70], radiation [91], ultrasound [92], and so on [93]. BMNs have a relatively mature application in disease diagnosis and environmental monitoring. However, in this review, we will focus on the use of enzyme-like activities of BMNs in cancer therapy.

In cancer treatment, nanozymes with CAT, POD, glucose oxidase (GOx), and glutathione peroxidase (GSH) activities are commonly used [94,95]. Recently, RuCu NPs with dual enzyme-like activities were used as multifunctional nano-formulations [96]. As shown in Figure 5a,b, RuCu NPs performed POD-like activity and CAT-like activity in the tumor microenvironment, in which RuCu NPs could relieve tumor hypoxia by reacting with the overexpressed H_2_O_2_ to generate O_2_ and kill tumors by catalyzing H_2_O_2_ to produce highly toxic ·OH. After that, density functional theory (DFT) calculations were conducted to explore the mechanism of the BMNs’ catalytic reaction. Cu doping created additional band gaps around the Fermi energy level, and more energy bands near the Fermi level of RuCu NPs caused stronger electrochemical activities. Analogously, Wang and co-workers reported an Rh-based core–shell nanosystem (Au@Rh-ICG-CM) exhibiting CAT-like activity and used for enhancing PDT (Figure 5c) [70]. As presented in Figure 5d, compared with Rh NPs and Au@Pt nanostructures, Au@Rh performed superior CAT-like activity because of the porous Rh shell. The enzyme-like activity of BMNs was also shown in Zhou’s experiments, they created a CAT-mimicking nanozyme, PdPt@GOx/IR780 (PGI), that can catalyze H_2_O_2_ decomposition into ^•^O_2_^−^ under the ultrasound (US) stimulation (Figure 5e) [97]. It can be seen in Figure 5f,g that there is an obvious generation of ROS under US irradiation. These preliminary findings indicated that PGI NPs had good potential as sonosensitizers for sonodynamic therapy (SDT). In conclusion, the enzyme-like activity of BMNs has great potential for application in the treatment of diseases.

### 3.4. Stability

Nanomaterials are inherently unstable due to their extremely high specific surface area, surface energy, and interfacial energy. However, BMNs can be synthesized to improve stability. BMNs with stability can be obtained by adjusting the composition, structure, and shape of the two metal elements, referring to the method of improving stability by forming alloys in traditional metals [98].

The tendency of bimetallic mixing has been calculated by computer simulations, thus acquiring the most stable structure [99,100]. In Dean’s study, a database of 5454 low-energy bimetallic NPs was generated and analyzed, thus they identified thermodynamically stable NPs [101]. They investigated the alloys Au–Cu, Ag–Au, and Ag–Cu with the structure of icosahedrons (Ico), cuboctahedrons (Cub), and elongated pentagonal bipyramids (EPBs). Overall, they reported minimum-energy configurations and illustrated several typical structures in Figure 6a. The plots for a bimetallic NP made of elements A and B were created, where each bond type (A–A, B–B, and A–B) was calculated. Through thermodynamic calculation resulting in Figure 6b, it is concluded that Ag–Au and Au–Cu NPs are more stable when the mixing degree is high, while Ag–Cu NPs perform unfavorable mixing behavior. This does not mean that Ag and Cu cannot form BMNs, but guides them to minimize the number of heteroatomic bonds via forming core–shell structure. In this way, the formation of BMNs in alloys or core–shell structures could better maintain the stability of nanomaterials.

The stability of BMNs is also reflected in heat resistance, electrode stability, colloidal stability, and other aspects, which have important applications in electrochemistry, disease diagnosis, and treatment. Xu et al. reported the element segregation characteristics and thermal stability of Ni–Pd bimetallic NPs [102]. In this study, it is confirmed that the catalytic activity and carbon deposition resistance of the Pd_x_Ni_1−x_ NP catalyst can be improved by doping Pd into Ni NP, while much more dopped-Pd atoms will gradually reduce sintering resistance. Huang et al. modeled six different Pt–Co NPs and compared their potential energies, as shown in Figure 6c [103]. The pure Pt and Co NPs are also measured as a comparison. It can be found in Figure 6d that the potential energy of Co is the highest, indicating that Co has the least stability and the commixture of Pt–Co greatly improves the stability of BMNs by reducing potential energy. For both Pt_3_–Co and Pt–Co NPs, the potential energy of ordered intermetallic is lower than that of disordered alloy with the same composition, indicating that the ordered intermetallic has better stability. Moreover, the Co@Pt core–shell structure is the least stable among the three types of configurations because that possesses higher energy than other Pt–Co nanoparticles with the same composition.

**Figure 6 molecules-27-08712-f006:**
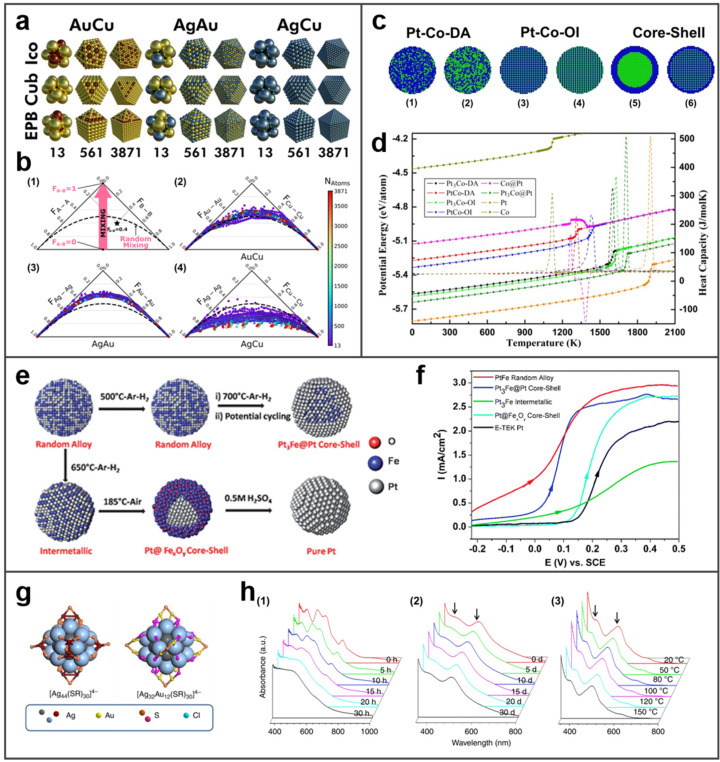
The stability of BMNs. (**a**) The most stable chemical ordering and composition at the given size, shape, and metal pair. Reprinted with permission from Ref. [101]. (**b**) Bond composition plots: (1) guiding plot illustrating the theoretical mixing limits from no heteroatomic bonds (F_A−B_ = 0) to no homoatomic bonds (F_A−B_ = 1); (2) (3) and (4) the preferential compositions of AuCu, AgAu, and AgCu NPs at all sizes and structures in thermodynamics. Reprinted with permission from Ref. [101]. Copyright © 2020 American Chemical Society. (**c**) Schematic illustration of Pt–Co bimetallic NPs: (1) disordered Pt_3_–Co alloy (Pt_3_Co-DA), (2) disordered Pt–Co alloy (Pt–Co–DA), (3) ordered Pt_3_Co L12 intermetallics (Pt_3_Co–OI), (4) ordered PtCo L10 intermetallics (PtCo-OI), (5) Co–Pt core–shell structure (Co@Pt), and (6) Pt_3_Co–Pt core–shell structure (Pt_3_Co@Pt). Coloring denotes the type of atom: blue, Pt atom; green, Co atom. Adapted with permission from Ref. [103]. (**d**) Temperature dependence of potential energy and specific heat capacity for different Pt–Co bimetallic NPs (the dashed lines correspond to the heat capacity). Reprinted with permission from Ref. [103]. Copyright © 2017 American Chemical Society. (**e**) Pt–Fe bimetallic NPs with different architectures. Reprinted with permission from Ref. [104]. (**f**) Polarization curves on different Pt-Fe catalysts. Reprinted with permission from Ref. [104]. Copyright © The Royal Society of Chemistry 2011. (**g**) Structures of Ag NCs and Ag@Au NCs (the hydrocarbon tails and carboxylic groups of the protecting ligands are omitted). Reprinted with permission from Ref. [105]. (**h**) UV absorption spectra of (1) [Ag_44_(SR)_30_]^4−^ and (2), (3) [Ag_32_Au_12_(SR)_30_]^4−^ NCs (The arrows indicate the absorption features at 390 and 490 nm). Reprinted with permission from Ref. [105]. Copyright © 2022 Springer Nature Limited.

Furthermore, the formation of BMNs is of great help to the recycling of electrocatalytic properties. Pt–Fe bimetallic NPs with various architectures (Pt–Fe random alloys, Pt_3_Fe@Pt core–shells, Pt_3_Fe intermetallics, and Pt@Fe_x_O_y_ core–shell−like systems, as shown in Figure 6e) have been reported in Liu’s study [104]. Figure 6f shows that the Pt–Fe BMNs perform great electrochemical properties. To their preliminary studies, the Pt–Fe random alloy, Pt_3_Fe intermetallic, and Pt_3_Fe@Pt core–shell NPs are stable in H_2_SO_4_ electrolyte, and there is no degradation in their catalytic performance after 500 potential cycles. However, Pt@FexOycatalyst that only has a single metal as a core shows poor stability. What is more, Yao et al. greatly improved the stability of Ag@Au nanoclusters (NCs) by doping Au heteroatom into the core of [Ag_25_(SR)_18_]^−^ (SR denotes thiolate ligand) to form bimetallic NCs [105]. It can be seen from Figure 6h that the absorption peak of [Ag_44_(SR)_30_]^4−^ NCs changed significantly after a few hours, while [Ag_32_Au_12_(SR)_30_]^4−^ NCs were stable in aqueous solution over 30 days at 25 °C and did not metamorphose at high temperature.

Additionally, the colloidal stability of BMNs is crucial for nanomaterials in disease treatment. In Vareessh’s work, the platinum–gold bimetallic NPs (PtAuBNps) have been synthesized to achieve intelligent targeted drug delivery, which exhibited excellent physiochemical attributes and colloidal stability [106]. Similarly, in Klebowski’s paper, PtAu and PdAu nanocomplexes possess significant stability compared with Au NPs, Pd NPs, and Pt NPs, they also show superior therapeutic effects [64].

## 4. Application of BMNs in Cancer Therapy

BMNs have been widely used in biomedical fields, especially in biosensing and imaging, due to their unique catalytic and optical properties. In addition, many studies have shown that BMNs are also promising nanotechnology in cancer therapy. Here, we outline the applications of BMNs in drug/gene delivery, PTT, PDT, chemodynamic therapy (CDT), radiotherapy (RT), and multimodal synergistic therapy based on their properties.

### 4.1. Drug/Gene Delivery

The use of chemotherapy drugs is the most mature and wide modality for cancer treatment. Some pathogenic mechanisms and corresponding drugs have been almost studied for various cancers [107]. The introduction of nanotechnology has created conditions for the delivery of multifarious anticancer drugs and improved anticancer efficacy [108]. Among them, the delivery of drugs by NPs is the most common and effective nano−mediated chemotherapy. BMNs can be regarded as good drug carriers because of their variety of shapes, large surface area, and the special interface from two metals combination. A variety of anticancer drugs and target molecules can be attached to BMNs to achieve the effect of chemotherapy on tumors without harming the normal body. For example, the addition of folic acid (FA) and doxorubicin (DOX) to PtPd NPs can improve their overall chemotherapy efficacy [109]. In this experiment, PtPd−ethylene glycol-FA-DOX nanoparticles (PtPd−PEG−FA−DOX) were synthesized. DOX is efficiently loaded on PtPd NPs and released regularly; meanwhile, it can attack cancer cells under the guidance of the target molecule FA. Similarly, Wu et al. designed a multifunctional intelligent drug delivery system to combine targeting, adequate drug loading, and controllable drug release [110]. As shown in Figure 7a, Ag–Au bimetallic NP as the core was coated with poly(ethylene glycol) (PEG), and loaded surface hyaluronic acid (HA) chains as targeting ligands and temozolomide (TMZ). As illustrated in Figure 7b,c, more TMZ is released with the increase in temperature, and according to the detection results of cell viability, the NP with drug downloaded by NIR irradiation performance stronger toxicity. The NP has a high drug-loading capacity and can be targeted to tumor cells and intelligently release TMZ at specific temperatures.

In addition to loading small organic molecules as targeted ligands, the most commonly effective multifunctional delivery systems are the linking of DNA or RNA. In the study of Kang et al., a DNA−coated NIR light-responsive drug delivery platform was constructed based on Au–Ag NRs [111]. Au–Ag NR-based nanogels possess the function of specific tumor recognition by concentrating DNA (sgc8c), and achieve the function of temperature-controlled rapid release by loading DOX in the gel (Figure 7d–f). When exposed to NIR light, the drug is released rapidly; moreover, adding DOX-loaded sgc8c conjugated core–shell nanogels (DOX-loaded sgc8c-NGs), Au–Ag NRs can achieve better target tumor-killing effect, so as to achieve the targeted therapy effect that can be controlled remotely. What is more, loading with therapeutic suicide genes on the BMNs can directly control the death of tumor cells in a genetic program to achieve gene therapy. Dutta et al. bounded pDNA to Au–Ag bimetallic nanocluster to form a composite nanoparticle [112]. Subsequently, the therapeutic effect was shown to be the suicide gene-mediated killing of cancer cells.

In conclusion, BMNs as nanocarriers for drug/gene delivery have the advantages of good water solubility, high loading capacity, and stability compared with MMNs, increased drug accumulation efficiency at the tumor site, and enhanced drug resistance. Moreover, surface modification of BMNs allows for specific and spatiotemporally controlled release of drugs for on-demand chemotherapy. It is believed that the BMNs-based nano-delivery system will obtain more fully applied in the field besides chemotherapy and GT, and contribute to cancer treatment in the future.

### 4.2. Enzyme-Mediated Tumor Therapy

BMNs exhibit excellent enzyme-like activity, which can be used in cancer treatment, especially in starvation therapy and CDT. Starvation therapy involves starving diseased cells to death by depleting the substances they need to grow and reproduce, such as nutrients, energy, and oxygen. Glucose is the main energy provider for cell growth, so GOx has become the most commonly used enzyme for starvation treatment [113]. When GOx acts as a catalyst, it consumes both glucose and oxygen, producing gluconic acid and H_2_O_2_, as shown in Figure 8a. This catalytic process blocks energy and oxygen supply to tumor cells, starving them to death. The H_2_O_2_ can significantly increase the oxidative stress of tumors, and can also produce hydroxyl radicals to kill tumor cells under the light. In addition, the production of gluconic acid also enhances the acidity of the microenvironment around the tumor, which can initiate some pH response processes [114].

A novel enzymatic cascade nanoreactor Pd@Pt-GOx/HA with controllable enzymatic activities has been developed in Zheng and co-workers’ study [115]. As shown in Figure 8b, HA on the surface of Pd@Pt-GOx/HA can actively target tumor cells that overexpress CD44 and decomposes after phagocytosis to release internal Pd@Pt-GOx. After that, GOx consumes O_2_ and glucose to achieve starvation treatment. At the same time, the H_2_O_2_ and acidic environment stimulate the activity expression of Pd@Pt nanozyme, which then catalyzed the generation of abundant highly toxic ^•^OH degradated from H_2_O_2_, achieving the CDT. As validation, Figure 8c shows that after the treatments of Pd@Pt-GOx/HA, tumors are significantly reduced and the survival rate of tumor-bearing mice is greatly improved. This treatment strategy is also effective for some pathogenic bacteria. For example, Peng et al. designed an innovative bimetal metal-organic framework domino micro-reactor (BMOF-DMR), which can kill Gram-negative (Escherichia coli, *E. coli*) and Gram-positive (Staphylococcus aureus, *S. aureus*) bacteria (Figure 8d) [116]. The antibacterial effect is shown in Figure 8e, which suggested the BMNs perform superior antibacterial activity against both bacteria. In addition, there are compositions of BMNs themselves with several kinds of enzyme-like activities, such as the biomimetic CoO@AuPt nanozyme reported in Fu’s work [117]. This nanozyme can initiate a remarkable CDT in the tumor microenvironment without any external stimuli by catalyzing a cascade of reactions to produce ROS (Figure 8f). As proven in Figure 8g, the CoO@AuPt exhibited good glucose depletion activity, GSH consumption activity, catalase-mimic activity, and ROS generation capacity. The BMNs’ therapeutic effects have also been demonstrated in animal studies, where they can effectively eliminate tumors without damaging the body (Figure 8h).

According to these investigations, it can be seen that the BMNs-based enzyme-mediated CDT strategy shows a potent curative effect on tumors, which is believed to be a potential hotspot in future research.

### 4.3. Radiation Therapy

RT is one of the most standardized treatments for unresectable tumors. Radiation sensitization and X−ray computed tomography imaging resulting from X−ray absorption and action on the surface of BMNs can be significantly enhanced by changing BMNs’ structure and morphology [118]. The snowflake-like Au nanocarriers (S−AuNC) studied by Kim et al. promoted the death of immunogenic cells and induced the release of anti-programmed death ligand 1 (aPD−L1) under radiation [119]. According to the results of Figure 9b, for HMGB1 cells treated with both S-AuNC and RT, the cell apoptosis was significantly higher than that treated only with S−AuNC or RT. Meanwhile, it is obvious from Figure 9c that the expression of PD−L1 in the cells treated with both S−AuNC and RT is more tempestuous, which confirms that S−AuNC is equipped with the function of radiosensitization. Finally, in terms of the therapeutic effect on the tumor, the tumor treated by S−AuNC and RT did not increase significantly and obtained the best therapeutic effect (Figure 9d). Currently, many tumors unaccommodated for surgical resection are treated with RT, which involves radioactive laser toward tumors directly. However, there is a critical problem the ability of normal tissues around the tumor to withstand the radiation intensity limits the use of RT [120]. Therefore, it is very important to develop radiosensitizers with highly accurate targeting properties. Certainly, the tumor’s response to radiation can be enhanced and much more mechanisms of tumor elimination can be stimulated through radiation by combining other therapies.

### 4.4. Photodynamics Therapy

PDT is an important cancer therapy strategy that uses photosensitizers to catalyze the production of ROS from tissue oxygen under laser irradiation to kill tumor cells [121]. The photocatalytic properties of BMNs make them ideal photosensitizers for PDT. In the study of Bi et al., MnO_2_−Pt@Au_25_ nanosheets were established as photosensitizers to enhance PDT by reducing the concentration of reductive GSH at the tumor site [122]. The consumption of excessive intracellular GSH by MnO_2_−Pt@Au_25_ reduced the consumption of ^1^O_2_, thus improving the efficiency of PDT (Figure 10a). As proven in Figure 10b,c, the production of ROS increased with time, the growth of tumors treated with MnO_2_−Pt@Au_25_ was inhibited and the volume of tumors was reduced. BMNs used in PDT can also be achieved by photocatalysis of two metals, respectively. Li et al. encapsulated Au NPs and Ru NPs into dendritic mesoporous silica (DMSN−Au−Ru NPs) to obtain a cascade catalytic ability for enhancing PDT (Figure 10d–f) [123]. Au NPs with enzyme−like activity can produce H_2_O_2_, and Ru NPs can further catalyze the decomposition of H_2_O_2_ to produce toxic ^1^O_2_. Eventually, tumors treated with DMSN−Au−Ru NPs and laser showed promising results.

In addition to some BMNs with intrinsic photocatalytic properties being photosensitizers, BMNs can also connect with photosensitive molecules to achieve a photocatalytic effect. For instance, Wei et al. designed a photosensitizer−Pd@Pt nanosystem (Pd@Pt−PEG−Ce_6_) for highly efficient PDT, which effectively solved the aggravation of tumor hypoxia caused by the consumption of O_2_ in PDT [124]. As shown in Figure 10g, Pd@Pt−PEG−Ce_6_ is composed of Pd@Pt with catalase−like activity and photosensitizer Ce_6_, which can catalyze H_2_O_2_ to produce O_2_ under laser irradiation to relieve hypoxia of tumor and provide oxygen for PDT. Figure 10h implied a large production of ROS, and Figure 10i demonstrated that Pd@Pt−PEG−Ce_6_ visibly inhibited tumor growth and significantly enhanced PDT after 808 nm and 660 nm laser irradiation. Figure 10j revealed Mon et al.’s study, a 5−ALA/Au−Ag−PEG−Ab NC (nanoconjugate combined with PEG functionalized Au−Ag nanoparticles, 5−aminolevulinic acid, and antibodies) with specific targeting was established to achieve precise PDT on MCF−7 breast cancer cells [125]. The cells treated with both 5−ALA/Au−Ag−PEG−Ab NC and laser showed obvious deformation and separation in Figure 10k, indicating the superior PDT effect.

Apparently, PDT is the most typical application of BMNs’ photocatalytic and enzyme-like activity, which has shown excellent efficacy in experiments. In addition, the combination of PDT and other therapies will achieve more perfect results.

### 4.5. Photothermal Therapy

PTT, involves nanoparticles absorbing irradiated laser and converting optical energy into heat to ablate tumor cells. Almost all BMNs with photothermal conversion properties can be regarded as good photothermal agents for PTT. Li et al. prepared Au–Ag core–shell NPs as photothermal agents to achieve highly efficient (Figure 11a) [126]. Au-Ag core–shell NPs show excellent photothermal conversion ability, as shown in Figure 9b, the temperature increases rapidly under laser irradiation; furthermore, in the test of cytotoxicity in Figure 11c, cells treated with the NPs and laser irradiation died acutely, indicating excellent PTT.

Bimetallic photothermal agents can achieve better photothermal conversion than single plasmon resonance metal NPs, which was demonstrated by much work. As an example, Au NRs coated with a shell of Pt nanodots (Au@Pt nanostructures) were proved to have a better photothermal effect than Au NRs in Tang and co-workers’ study [127]. When Au@Pt nanostructures and Au NRs were irradiated with laser for the same power and time, the temperature of Au@Pt nanostructures group increased faster and higher, and the cells treated with Au@Pt nanostructures were ablated more than that treated with Au NRs. At the same time, Au@Pt nanostructures also showed superior PTT than Au NRs in tumors of mice (Figure 11d). Another BMNs hollow Au–Ag alloy nanourchins (HAAA-NUs) can also demonstrate it [128]. The NUs resemble a “sea urchin”-like shape in Figure 11e. HAAA-NUs showed better photothermal conversion ability than Au colloids (Figure 11f). What is more, HAAA-NUs perform good biocompatibility; they show the ability to kill cells only under laser irradiation, and the damage to cells increases with the laser’s power. It is implied from the images that the tumor of the mice injected with HAAA-NUs (Figure 11g, bottom rows) showed strong ablation after laser irradiation, while the temperature of that without injection of HAAA-NUs (Figure 11g, top rows) did not increase significantly. Surprisingly, the tumors injected with HAAA-NUs and irradiated with laser significantly diminished and were eventually completely eliminated (Figure 11h). Coincidentally, ceria-loaded gold@platinum nanospheres modified with PEG (CeO_2_/Au@Pt-PEG) enabled superior and targeted PTT (Figure 11i) [129]. The results in Figure 11j show the superior photothermal properties of CeO_2_/Au@Pt-PEG in aqueous solution, in vitro and in vivo, and confirm the obvious therapeutic effect of CeO_2_/Au@Pt-PEG on tumors.

### 4.6. Synergistic Therapy

Emerging nano-mediated therapies may offer interesting alternatives to traditional surgical treatment. There are a number of studies that have shown that combinations of treatments are much more effective than single treatments [130]. The simplest and easiest synergistic therapy is a combination of two treatments, especially the combination of PTT or PDT with another therapy. It can be easily achieved because of the optical properties of BMNs. In the study of Liu et al., a BMN with two enzyme-like activities was established, which provides the possibility to realize the combination treatment of PDT and starvation therapy [12]. As we can see from Figure 12a, porphyrin metal−organic frameworks (PCN) are used as carriers to adulterate CAT−mimicking Pt NPs and GOx−mimicking Au nanoparticles at different layers, separately. Finally, a targeted FA is attached to the BMNs’ surface to form P@Pt@P−Au−FA. The P@Pt@P−Au−FA as dual−nanozymes can catalyze the production of O_2_ by decomposing the H_2_O_2_ produced in the oxidation of glucose. The produced O_2_ can not only promote the catalysis of glucose, thus promoting starvation treatment, but also assist the production of ROS in PDT. The tumors treated with P@Pt@P−Au−FA and laser showed obvious reduction and elimination, proving the superior therapeutic effect on tumors (Figure 12b). Meanwhile, the treated mice continued to grow normally while eliminating the tumor, indicating there is no harm to the normal body (Figure 12c). Homoplastically, an ultrathin single−site bimetallic (copper hexacyanocobaltate) nanosheet (CuCo NS) has been reported and shown in Figure 12d, which can achieve better synergistic treatment by enhancing the production of ROS in virtue of PTT [131]. In addition, CuCo NS showed excellent therapeutic effects against tumors (Figure 12e).

In order to achieve more brilliant therapeutic, collaborative therapy comprised consist of various treatments has mushroomed. Hollow mesoporous organotantalum nanospheres modified with Au and Pt dual nanoenzymes (HMOTP@Pt@Au@DOX), for instance in Figure 13a, were designed to achieve the synergistic therapy of RT, chemotherapy, and starvation treatment [132]. The HMOTP@Pt@Au@DOX presented chemo−radio sensitization and could be self−activated under GSH stimulation to release DOX on−demand. Moreover, the BMNs as nanoenzyme triggered cascade catalytic reactions under radiation, completing the cycle of consuming glucose and O_2_, and replenishing O_2_; producing ROS to kill cancer cells at the same time. Analogously, Au@Pt−DOX−PCM−PEG nanotherapeutics (adding DOX into the mesopores of Au@Pt nanostructures assisted by phase change materials) are reported by Sun et al. that combined chemotherapy, PTT, and GT [133]. The efficiency of PTT was improved by alleviating tumor hypoxia and reducing tumor heat tolerance with gene therapy in this study; additionally, genic−enhanced PTT combined with chemotherapy achieved significant enhancement (Figure 13b,d).

In addition, using BMNs as biosensors has realized the detection of tumor biomarkers and living tumor cells, which has solved the problem of early diagnosis of cancer and has been well−developed [134,135]. Ding et al. reported an ultrasensitive cytosensor based on Cu_2_O@PtPd, in which accurate recognition of cancer markers and monitoring of drug concentration can demonstrate the effect of tumor therapy, and further targeted therapy can be conducted according to the results [136]. Meanwhile, the performance of BMNs in biological imaging is also very attractive. Therefore, the strategy of integrating diagnosis and treatment used BMNs becomes particularly significant. For example, collaborative therapy and imaging can be combined to achieve visual precision treatment and achieve the integration of diagnosis and therapy. The cRGD−modified the DOX−loaded Au@Pt NPs (DOX/Au@Pt−cRGD) can not only achieve excellent chemo-photothermal co-therapy without damaging normal tissues but also realize fluorescence imaging by the optical properties of BMNs (Figure 13c,e) [137]. Another classic example is the core–shell Au@Cu_3_(BTC)_2_ NPs designed for multifunctional Raman imaging and chemo-phototherapy [138]. The Au@Cu_3_(BTC)_2_ NPs exhibited high stability in the complex physical environment and performed Raman and NIR dual-responsive, hence regarded as potential theranostic nanoplatforms.

In a word, using BMNs as nanocarriers to achieve chemotherapy and GT is the most basic application, and there is no obvious advantage of therapeutic effect over traditional chemotherapy. The starvation treatment and CDT mediated by the enzyme-like activity are novel and can achieve better results by combining with other therapies. In addition, BMNs as radiosensitizers, photothermal agents, and photosensitizers are the most concerned applications in nanotechnology−mediated tumor therapy, which have demonstrated remarkable therapeutic effects. Moreover, BMNs-based multimodal cancer therapy can integrate the advantages of different therapies to achieve better therapeutic effects, which heralds a new era of synergistic therapy.

## 5. Conclusions and Prospects

There is no doubt that BMNs possess many advantages, such as exceptional catalytic activity and optical properties, which make them quickly develop into an ideal therapeutic option in cancer therapy. Especially, the attractive activities of BMNs are due to the synergistic effects between different metals, which make them have broader applications than MMNs. However, thoroughly exploring the growth, catalytic, and pharmacokinetics mechanism of BMNs and building reliable clinical devices are still urgently needed to be addressed. Based on the above situations, this review summarizes the synthesis methods, properties, and practical applications of BMNs associated with the treatment of tumors. The main motivation for the study comes from the remarkable properties, including complex morphologies, diverse enzyme-like activities, superior stability, and perfect optical properties. In addition, BMNs truly achieve the effect that one plus one is greater than two, and the properties of the two metals can be enhanced by each other. Thanks to the exploration and research of BMNs by many scholars, we can put forward conclusions and suggestions in the medical field, especially in the treatment of cancer for the future.

First of all, it is still challenging to find suitable synthetic methods from many methods that can be used to produce BMNs for cancer therapy. It has been found that BMNs with zero dimensions, such as spheres, rods, stars, and flowers, are usually used in cancer treatment. These structures are usually synthesized by seed-mediated and hydrothermal methods, where the regulation of BMNs’ morphology has been well studied. In addition, we suggest that the growth pattern and mechanisms of all BMNs can be thoroughly explored by molecular simulation to establish a material library so that nanostructures with specific properties and shapes can be designed more clearly to reply to complex environments and various therapeutic applications. 

Secondly, BMNs have extensive applications in cancer therapy, but there are still many properties that remain unexplored. Many BMNs show good catalytic activity and high sensitivity that can act as enzymes in the tumor microenvironment. However, most of the current studies on the enzyme-like activities of BMNs are catalase, GOx, and glutathione peroxidase, and the development of BMNs with more enzyme-like activities would be beneficial to further expand its application in biological fields. It is still a big challenge at the moment, but it is fascinating and worth solemn attention.

Thirdly, BMNs have been shining in synergistic therapy and made considerable achievements. In view of the mature capabilities of BMNs in intelligent drug delivery and cancer diagnosis, we suggest that using BMNs as smart nanoplatforms creates more diversified treatment strategies as well as automatically assesses the status of the tumor to adjust the level of treatment. Therefore, it is of great significance to make full use of BMNs’ sensing and imaging capabilities, loading capacity, catalytic performance, and photothermal properties to construct a multifunctional intelligent nanosystem.

Last but not least, it is crucial to establish an effective and authoritative evaluation system to judge the therapeutic effect of BMNs. In the investigation, it is found that no matter which nanocarrier is used for tumor treatment, it only represents the volume and weight of tumors and the weight of mice. What is a veritably “effective” treatment in clinical application, this is a worthy direction for all medical material researchers and physicians to work together.

## Figures and Tables

**Figure 3 molecules-27-08712-f003:**
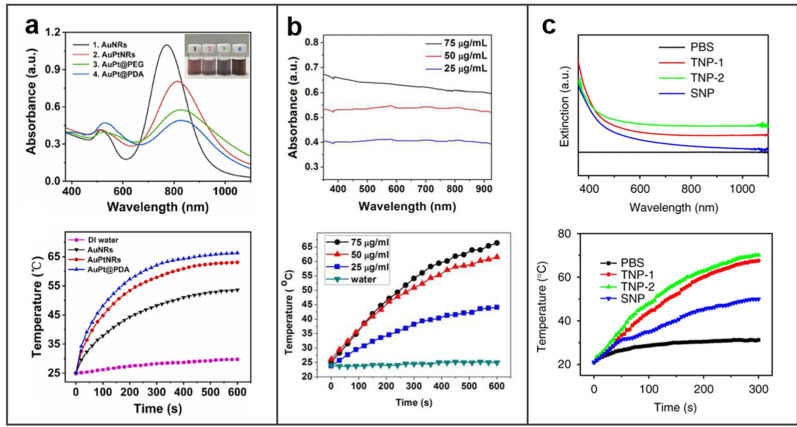
Photothermal effect of BMNs. (**a**) UV/vis absorption spectra and temperature curves of NPs. Reprinted with permission from Ref. [74]. Copyright © 2022 Elsevier B.V. (**b**) UV/vis-NIR absorption spectra and photothermal effects of the aqueous dispersion of PEGylated Au@Pt NDs. Reprinted with permission from Ref. [62]. Copyright © 2016 American Chemical Society. (**c**) UV/vis spectrum and temperature changes of CuPd NPs in aqueous solution. Reprinted with permission from Ref. [63]. Copyright © 2022 Springer Nature Limited.

**Figure 4 molecules-27-08712-f004:**
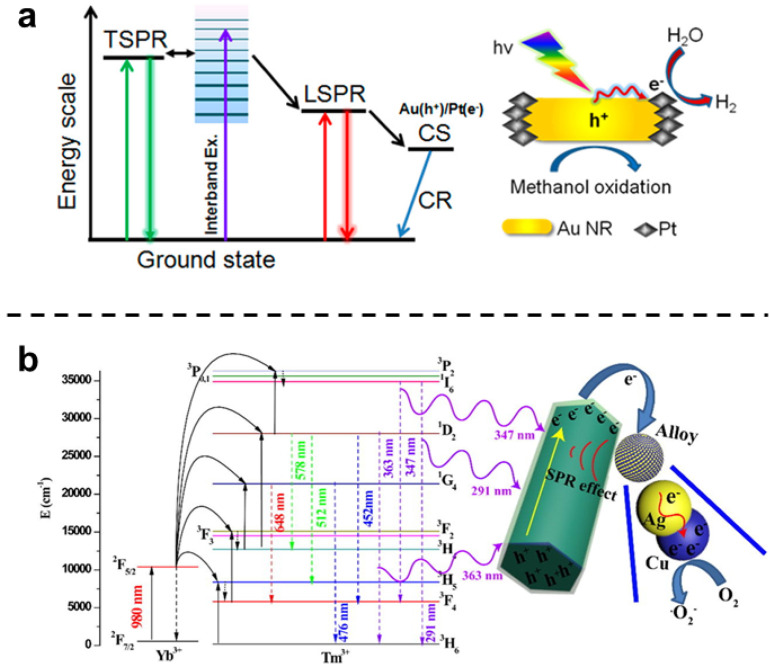
Photocatalytic mechanism of BMNs. (**a**) Schematic diagram for radiative decay of surface plasmon (left) and reaction mechanism for H_2_ production (right) in Pt-modified Au nanorods (NRs). Reprinted with permission from Ref. [77]. Copyright © 2014 American Chemical Society. (**b**) Possible mechanism for photocatalytic disinfection and the process of generating reactive radical ^•^O_2_^−^. Reprinted with permission from Ref. [78]. Copyright © 2018 Elsevier Inc.

**Figure 5 molecules-27-08712-f005:**
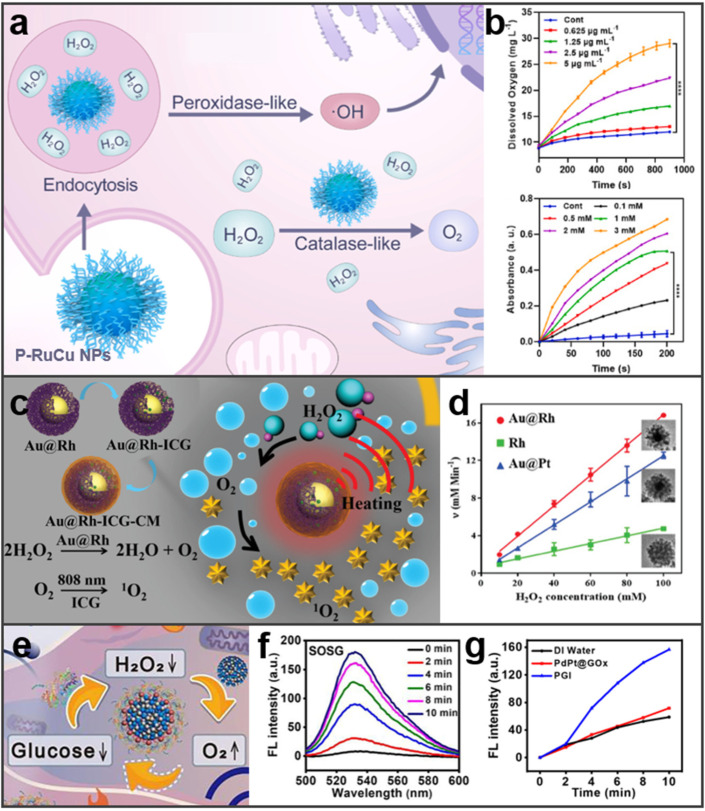
Expression of enzyme-like activity of BMNs. (**a**) Scheme illustration of RuCu NPs for enhanced cancer radiotherapy. Reprinted with permission from Ref. [96]. (**b**) Dual enzyme-like activities of RuCu NPs. Reprinted with permission from Ref. [96]. Copyright © 2022 Elsevier Ltd. (**c**) The associated major mechanistic pathways of Au@Rh ICG-CM in cancer therapy. Adapted with permission from Ref. [70]. (**d**) The velocity of the catalytic reaction. Reprinted with permission from Ref. [70]. Copyright © 2020 John Wiley & Sons, Inc. (**e**) Scheme diagram to show the catalytic activity of PGI. Reprinted with permission from Ref. [97]. (**f**) Fluorescence spectra of PGI NP dispersion containing SOSG. Reprinted with permission from Ref. [97]. (**g**) Fluorescence emission intensity under US irradiation. Reprinted with permission from Ref. [97]. Copyright © 2022 Elsevier B.V. **** *p* < 0.0001.

**Figure 7 molecules-27-08712-f007:**
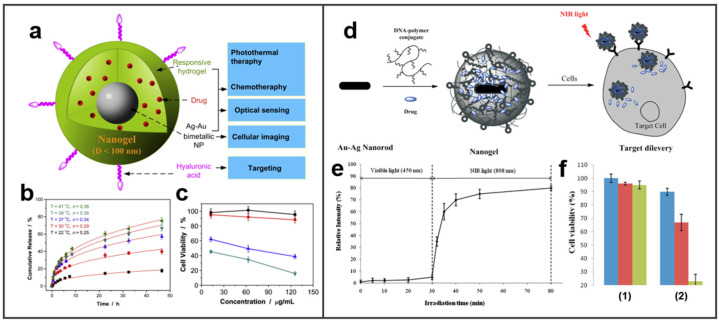
BMNs are used as drug carriers for tumor chemotherapy. (**a**) Schematic illustration of multifunctional Ag–Au core–shell hybrid nanogels. Reprinted with permission from Ref. [110]. (**b**) Releasing profiles of TMZ from the PSG hybrid nanogels at different temperatures. Reprinted with permission from Ref. [110]. (**c**) Comparison of B16F10 cell viability following different treatments with PSG as drug carriers (■: empty hybrid nanogels; ●: empty hybrid nanogels with 5 min initial NIR irradiation; ▲: TMZ-loaded hybrid nanogels; ▼: TMZ-loaded hybrid nanogels with 5 min initial NIR irradiation). Reprinted with permission from Ref. [110]. Copyright © 2010 Elsevier Ltd. (**d**) Schematic of core–shell nanogels for targeted drug delivery. Reprinted with permission from Ref. [111]. (**e**) Controlled release of the entrapped fluorescein from the core–shell nanogels (10 nM) by visible and NIR light irradiations. Reprinted with permission from Ref. [111]. (**f**) Cytotoxicity assays of CCRF-CEM (target) cells in the absence of nanogels (1) and those incubated with (2) Dox-loaded sgc8c-NGs (2 nM), at 808 nm laser for 0, 5, and 10 min, respectively (blue, red, and green). Adapted with permission from Ref. [111]. Copyright © 2011 American Chemical Society.

**Figure 8 molecules-27-08712-f008:**
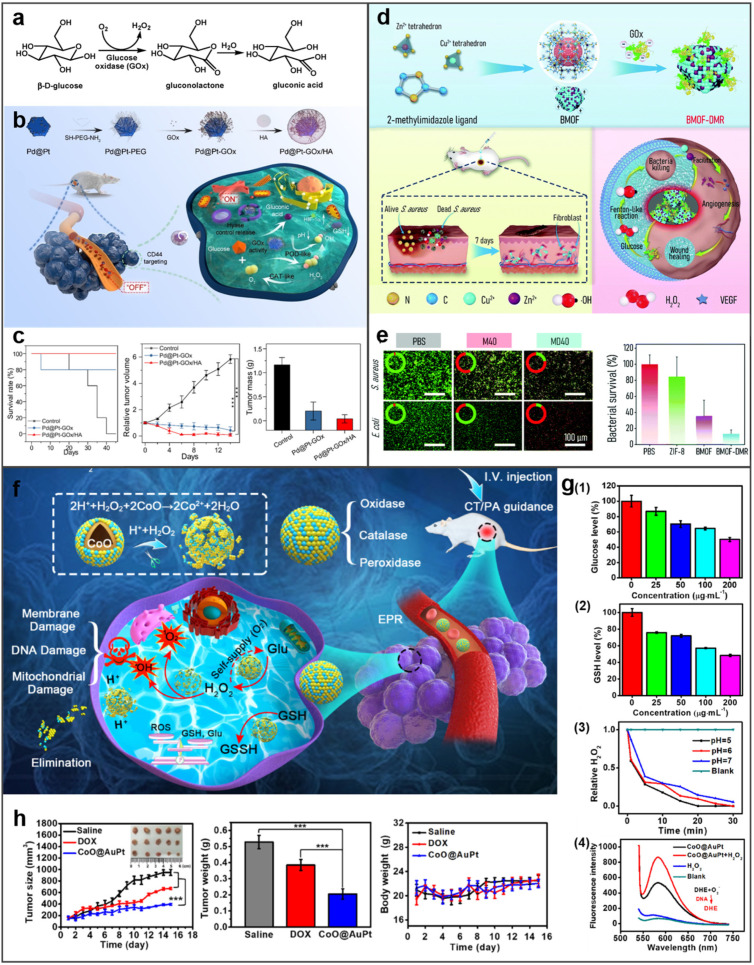
The applications of BMNs in starvation therapy and radiation therapy. (**a**) Mechanism of action of glucose oxidase. (**b**) Nanozyme-mediated starving-enhanced CDT of the Pd@Pt-GOx/HA nanoreactors. Reprinted with permission from Ref. [115]. (**c**) Antitumor effects through the nanozyme-mediated starving-enhanced cancer therapy in the 4T1 model (survival rates, tumor growth curves, and tumor mass). Reprinted with permission from Ref. [115]. Copyright © 2020 American Chemical Society. (**d**) Therapeutic effect toward bacteria-induced wound healing and killing bacteria by the BMOF-DMR. Reprinted with permission from Ref. [116]. (**e**) The antibacterial effect of BMOF-DMR in the wound (fluorescence images of live/dead staining of *S. aureus* and *E. coli*; the surviving bacteria rates). Reprinted with permission from Ref. [116]. Copyright © Royal Society of Chemistry 2022. (**f**) Schematic illustration of CoO@AuPt NPs working for augmented CDT. Reprinted with permission from Ref. [117]. (**g**) The enzyme-like activity of CoO@AuPt. Adapted with permission from Ref. [117]. (**h**) Therapeutic effect of CoO@AuPt. Reprinted with permission from Ref. [117]. Copyright © 2022 Elsevier B.V. *** *p* < 0.001.

**Figure 9 molecules-27-08712-f009:**
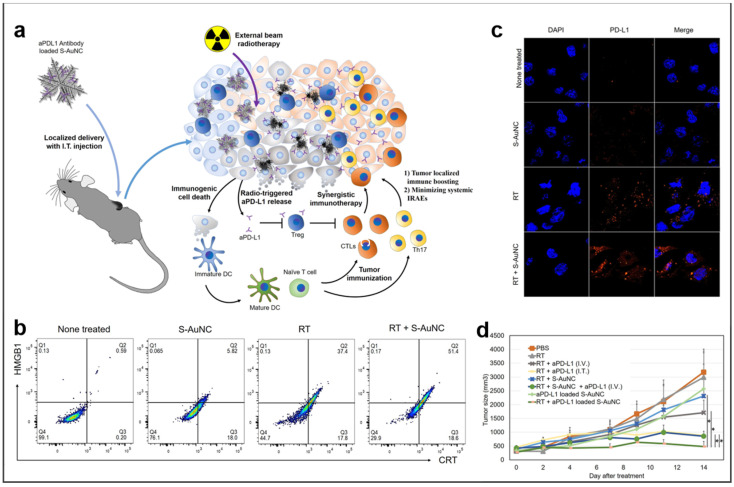
BMNs used in cancer radiotherapy. Reprinted with permission from Ref. [119]. (**a**) Schematic illustration of the radiation using radiation−responsive splintery metallic nanocarriers. (**b**) The ICD population of the group treated with S−AuNC, RT, and RT + S−AuNC. (**c**) Enhanced PD−L1 expression of Tramp C1 in response to the radiosensitization with S−AuNC. (**d**) Tumor growth graph. Copyright © 2020 American Chemical Society. * *p* < 0.05.

**Figure 10 molecules-27-08712-f010:**
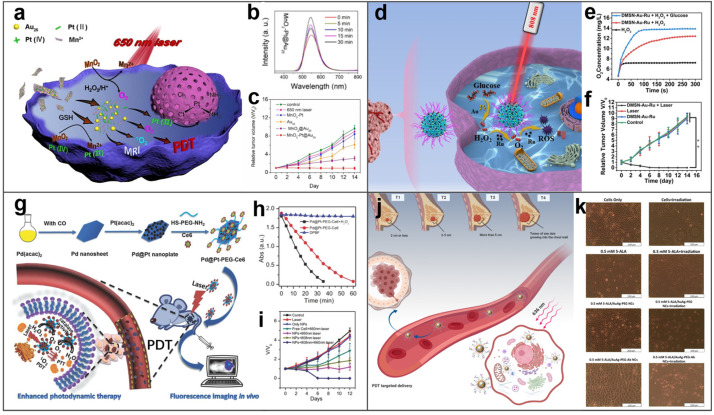
BMNs used in photodynamic therapy of tumors. (**a**) Scheme of photodynamic and chemotherapy based on biodegradable MnO_2_−Pt@Au_25_ nanosheets. Reprinted with permission from Ref. [122]. (**b**) Fluorescence spectra of DCFH solutions treated with MnO_2_−Pt@Au_25_. Reprinted with permission from Ref. [122]. (**c**) The relative tumor volume was normalized to the initial volume before treatment. Reprinted with permission from Ref. [122]. Copyright © 2022 Elsevier B.V. (**d**) Bimetallic nanozymes achieve synergistic effects of PTT and self−enhanced PDT. Reprinted with permission from Ref. [123]. (**e**) O_2_ generation in different conditions. Reprinted with permission from Ref. [123]. (**f**) V/V_0_ values of Kunming tumor−bearing mice within 14 days. Reprinted with permission from Ref. [123]. Copyright © 2017 John Wiley & Sons, Inc. (**g**) Scheme showing the preparation and application of Pd@Pt−PEG−Ce_6_. Reprinted with permission from Ref. [124]. (**h**) The ^1^O_2_ production efficiency of Pd@Pt−PEG−Ce_6_ after different times of irradiation. Reprinted with permission from Ref. [124]. (**i**) Changes in tumor volumes of mice treated in different ways. Reprinted with permission from Ref. [124]. Copyright © 1999–2018 John Wiley & Sons, Inc. (**j**) The targeted PDT for breast cancer. Reprinted with permission from Ref. [125]. (**k**) Light microscopy images for PDT−treated and non−treated MCF−7 cells with free 5−ALA, the NCs, and laser light irradiation under the IC50 concentration of 5−ALA at a wavelength of 636 nm and fluency of 5 J/cm^2^. Reprinted with permission from Ref. [125]. Copyright © 1996–2022 MDPI (Basel, Switzerland). ** *p* < 0.01.

**Figure 11 molecules-27-08712-f011:**
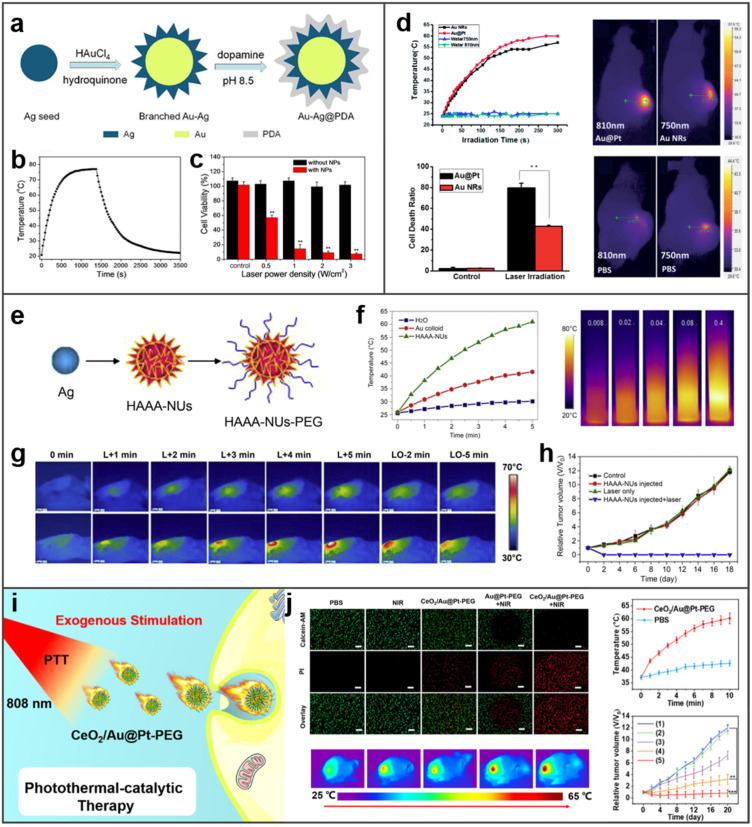
BMNs for photothermal therapy. (**a**) Schematic diagram of composition of Au–Ag@PDA. Reprinted with permission from Ref. [126]. (**b**) The temporal temperature variation of the branched Au–Ag NPs. Reprinted with permission from Ref. [126]. (**c**) Hela cell viabilities after 808 nm laser irradiation with different power densities for 10 min. Reprinted with permission from Ref. [126]. Copyright © 2015 American Chemical Society. (**d**) The photothermal therapeutic of Au@Pt nanostructures. Reprinted with permission from Ref. [127]. Copyright © The Royal Society of Chemistry 2014. (**e**) Strategy for HAAA-NUs synthesis and functionalization. Reprinted with permission from Ref. [128]. (**f**) Photothermal effect of HAAA-NUs. Reprinted with permission from Ref. [128]. (**g**,**h**) In vivo photothermal tumor therapy ((**g**) IR thermal images of tumor-bearing mice with laser irradiation, L-laser on, LO-laser off. (**h**) The tumor volumes after different treatments). Reprinted with permission from Ref. [128]. Copyright © 2014 Elsevier Ltd. (**i**) Synergistic photothermal-catalytic therapy of CeO_2_/Au@Pt-PEG. Reprinted with permission from Ref. [129]. (**j**) The therapeutic effect of CeO_2_/Au@Pt-PEG on tumors in vitro and in vivo (Relative tumor volume: (1) PBS, (2) NIR, (3) CeO_2_/Au@Pt-PEG, (4) Au@Pt-PEG + NIR, (5) CeO_2_/Au@Pt-PEG + NIR). Reprinted with permission from Ref. [125]. Copyright © 2022 Elsevier B.V. ** *p* < 0.01, *** *p* < 0.001.

**Figure 12 molecules-27-08712-f012:**
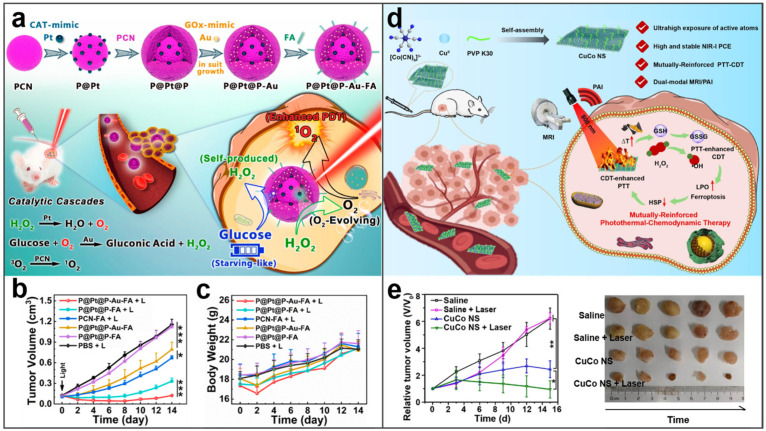
BMNs used in combination treatment with two therapies. (**a**) Schematic illustration of the catalytic cascades−enhanced synergistic cancer therapy. Reprinted with permission from Ref. [12]. (**b**) Tumor growth curves and (**c**) body weight changes of the 4T1 tumor−bearing mice in six groups. Reprinted with permission from Ref. [12]. Copyright © 2022 American Chemical Society. (**d**) Schematic illustration of the synthesis procedure for CuCo NS and proposed therapeutic mechanism of mutually reinforced PTT−CDT. Reprinted with permission from Ref. [131]. (**e**) The therapeutic effect of CuCo NS against tumors. Adapted with permission from Ref. [131]. Copyright © 2022 Elsevier B.V. * *p* < 0.05, ** *p* < 0.01, *** *p* < 0.001.

**Figure 13 molecules-27-08712-f013:**
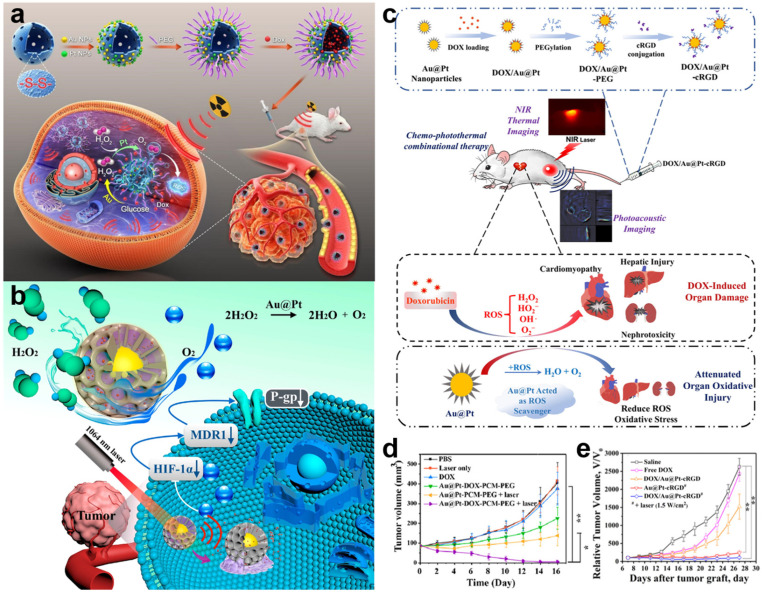
BMNs used in combination therapy. (**a**) Schematic diagram of the therapeutic function of HMOTP@Pt@Au@Dox. Reprinted with permission from Ref. [132]. Copyright © 2022 Elsevier B.V. (**b**) Schematic illustration of the key steps toward the synthesis of Au@Pt−DOX−PCM−PEG nanotherapeutic. Reprinted with permission from Ref. [133]. Copyright © 2022 American Chemical Society. (**c**) Schematic illustration of chemo-photothermal combinational therapy by the multifunctional Au@Pt NPs. Reprinted with permission from Ref. [134]. Copyright © 2017 American Chemical Society. (**d**) Time-resolved size change of the MDA−MB−231 tumors in nude mice after various treatments. Reprinted with permission from Ref. [133]. Copyright © 2022 American Chemical Society. (**e**) Relative tumor volume variation of the mice in different groups during treatment. Reprinted with permission from Ref. [134]. Copyright © 2017 American Chemical Society. * *p* < 0.05, ** *p* < 0.01.

**Table 1 molecules-27-08712-t001:** BMNs with photocatalytic properties, suitable laser, and products for application.

Types of BMNS	Products	Lasers	Ref.
Pt−modified Au NPs	H_2_	460 < λ < 820 nm	[77]
Ag–Cu alloy NPs	^•^OH radical, ^•^O_2_^−^ radical	λ = 980 nm	[78]
Au–Pt alloys	H_2_	λ = 325 nm	[79]
Pt@MIL−125/Au	H_2_	380 < λ < 800 nm	[80]
Au–Cu/CaIn_2_S_4_ composites	H_2_	420 nm ≤ λ ≤ 750 nm	[81]
Au/Pd/TiO_2_ nanoparticles	O^•−^ radical	310 < λ < 380 nm	[82]
Au–Pd/TiO_2_/NB nanostructures	O^•−^ radical	λ = 450 nm	[83]
Au@Pd@MOF−74	CO	λ = 707 nm	[84]
Pt/Au@Pd@MOF−74	CH_4_ (H^•^ radicals)	λ = 523 nm	[84]
Au@Ag/TiO_2_ NP	^•^OH radical, ^•^O_2_^−^ radical	λ = 664 nm	[85]
Cu–Au NPs	^•^O_2_^−^ radical	620 nm ≤ λ ≤ 670 nm	[86]
Ag/Au/TiO_2_ NPs	^•^O_2_^−^ radical	λ = 254 nm	[87]

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
