# Peer review of "Bimetallic Nanomaterials: A Promising Nanoplatform for Multimodal Cancer Therapy"

_molecules, 2022, doi:10.3390/molecules27248712_

Round 1

Reviewer 1 Report

In The Review article entitled “ Bimetallic nanomaterials: A promising platform for cancer therapy”, the authors systematically reviewed the various method of synthesis of Bimetallic nanomaterials and their application in multimodal cancer therapy. But authors some key applications of Bimetallic nanomaterial in cancer theranostics, a complete solution for diagnosis, treatment, and management of cancer. The article has many grammatical and sentence errors, and the language organization needs to be improved. For these reasons, I conclude that the paper should undergo minor revision

1.      The introduction is very general in nature. Authors need to more elaborate on the advantages and disadvantages of currently practiced therapeutic intervention, and how Bimetallic nanomaterials will help in overcoming their disadvantages.

2.      Authors need to add a section for the classification of Bimetallic nanomaterials based on their spatial arrangement such as complete alloy, shell-core, Yolk-shell, etc with examples of application in biomedicine.

3.  Authors need to add recent references on the application of Bimetallic nanomaterials in cancer in the last 3 years. Refer https://doi.org/10.1002/smll.202003496

https://doi.org/10.1016/j.biomaterials.2022.121811

https://doi.org/10.1016/j.molliq.2019.111303

https://doi.org/10.1002/elps.202100218

4. the application of Bimetallic nanomaterial in cancer theranostics is a key and important application, The authors need to add an elaborated section on the application of Bimetallic nanomaterial in cancer detection as well as cancer theranostics.

5. Conclusion needs to be improved by stating the advantages and disadvantages as well as their future scope with other emerging cancer treatment 

6.  Title seems very general. Please ensure all keywords are used in the title such as Multimodel therapy or cancer theranostics

7.      Typographical errors can be avoided. The language and grammar used throughout the manuscript need to be improved. 

Reviewer 2 Report

Thanks for the invitation to review this manuscript entitled: Bimetallic nanomaterials: A promising platform for cancer therapy. In my opinion, it’s an interesting topic. However, there are some issues that the author should address to improve the manuscript.

Because they discussed Bimetallic nanomaterials, the authors should be added more TEM images of NPs to show bimetallic NPs structures.

The authors also should be added some HR-TEM and EDS images of NPs to better show bimetallic NPs structures.

The quality of the Figures should be improved.

Studies that brought in section 3.3 Enzyme-like activity, are not related to the Heading. There is no enzymatic activity. For example in the case of RT, it is radioenhancing ability, not Enzyme-like activity.

In the same section, the authors don’t discuss properties, while it is one of the subheadings of 3. Unique properties of BMNs relevant to cancer therapy.

Abbreviations should be explained in the first-mentioned place. After that, it should be written as abbreviations throughout the text. 

For section 4 Application of BMNs in cancer therapy, the authors should be presented more and more studies and discussed and compared them.

Round 2

Reviewer 2 Report

Will be accepted